# Improving the specificity of nucleic acid detection with endonuclease-actuated degradation

Roger S. Zou [1,2], Momcilo Gavrilov [2,3], Yang Liu [2], Dominique Rasoloson[4,5], Madison Conte[6], Justin Hardick[6], Leo Shen [1,2], Siqi Chen[1,4], Andrew Pekosz [7], Geraldine Seydoux[4,5], Yukari C. Manabe[6] & Taekjip Ha [1,2,3,5 ✉]

Nucleic acid detection is essential for numerous biomedical applications, but often requires complex protocols and/or suffers false-positive readouts. Here, we describe SENTINEL, an approach that combines isothermal amplification with a sequence-specific degradation method to detect nucleic acids with high sensitivity and sequence-specificity. Target single-stranded RNA or double-stranded DNA molecules are amplified by loop-mediated isothermal amplification (LAMP) and subsequently degraded by the combined action of lambda exo-nuclease and a sequence-specific DNA endonuclease (e.g., Cas9). By combining the sensitivity of LAMP with the precision of DNA endonucleases, the protocol achieves attomolar limits of detection while differentiating between sequences that differ by only one or two base pairs. The protocol requires less than an hour to complete using a 65 °C heat block and fluorometer, and detects SARS-CoV-2 virus particles in human saliva and nasopharyngeal swabs with high sensitivity.

[1] Department of Biomedical Engineering, Johns Hopkins University School of Medicine, Baltimore, MD, USA. [2] Department of Biophysics and Biophysical Chemistry, Johns Hopkins University School of Medicine, Baltimore, MD, USA. [3] Department of Biophysics, Johns Hopkins University, Baltimore, MD, USA. [4] Department of Molecular Biology and Genetics, Johns Hopkins University School of Medicine, Baltimore, MD, USA. [5] Howard Hughes Medical Institute, Baltimore, MD, USA. [6] Division of Infectious Diseases, Department of Medicine, Johns Hopkins University School of Medicine, Baltimore, MD, USA. [7] Department of Molecular Microbiology and Immunology, Johns Hopkins University Bloomberg School of Public Health, Baltimore, MD, USA. ✉email: tjha@jhu.edu

Nucleic acid detection has a wide range of applications from epidemiology, disease diagnosis, and pathogen surveillance to laboratory use. While there are many methods for detecting nucleic acids, trade-offs exist between sensitivity, specificity, speed, cost, and convenience[1,2]. Development of new strategies not only expands the existing toolbox, but also may help resolve bottlenecks in diagnostic testing with significant public health implications, especially for the recent COVID-19 pandemic[3]. Quantitative PCR is the gold standard for nucleic acid detection but requires sophisticated equipment and dedicated laboratory space[4]. Isothermal amplification methods such as loop-mediated isothermal amplification (LAMP)[5], which uses a single heating temperature for target sequence amplification, are an alternate approach that avoids the use of specialized equipment but may suffer from lower specificity and false-positive readouts[6–12]. Recently, endonucleases from the clustered regularly interspaced short palindromic repeats (CRISPR) and CRISPR-associated (CRISPR-Cas) adaptive immune systems have been deployed for nucleic acid detection as an additional step on top of initial isothermal amplification to improve the specificity of target sequence recognition[13,14]. The CRISPR nuclease recognizes the target sequence in the amplified product, which catalyzes the cleavage of quenched fluorescent reporters to generate the detection signal. However, this approach has been limited to Cas12 and Cas13 due to the requirement for non-sequence specific collateral cleavage activity upon target sequence recognition. We sought to develop a more universal strategy that could be used with any sequence-specific endonuclease.

Sequence-specific endonucleases such as restriction enzymes or RNA-guided endonucleases (i.e., Cas9) are highly efficient and specific. Cleavage generates two new free 5′ ends which can be targeted by lambda exonuclease ($\lambda$-exo)[15], a highly processive 5′–3′ exonuclease, for rapid degradation. The initial isothermal amplification (using primers resistant to exonuclease activity) and subsequent degradation after endonuclease cleavage are monitored using convenient fluorescence-based measurements of dsDNA concentration. Samples with the expected target would exhibit an amplification signal after LAMP and a loss of signal after endonuclease and $\lambda$-exo digestion. Samples with the incorrect target would either fail to amplify with LAMP, resulting in minimal initial signal, or generate a LAMP amplification product resistant to endonuclease and $\lambda$-exo treatment. We term this new rapid nucleic acid detection platform Specific Enzymatic Nucleic Acid Targeting with Nucleases and Exonuclease Lambda (SENTINEL). We demonstrate that this approach combines the attomolar-level sensitivity of isothermal amplification with the nucleotide-level specificity of sequence-specific endonucleases. The SENTINEL assay is highly convenient in requiring minimal laboratory reagents, modular in the choice of sequence-specific endonuclease, fast with under 1-h total time and minimal hands-on time, and versatile in detecting SARS-CoV-2 virus particles in human saliva and nasopharyngeal swabs with high sensitivity.

## Results

For initial validation, we evaluated wild type SpCas9 as the endonuclease[16]. Using in vitro transcription, we generated synthetic RNA from the N-gene of SARS-CoV-2 as the target of interest[17] (Fig. 1a; Supplementary Fig. 1a). Reverse transcription LAMP (RT-LAMP) was first performed at 65 °C for 30 min. The RT-LAMP primers were designed to all contain 5′ phosphorothiolate modifications, resulting in product DNA that resists $\lambda$-exo degradation. Agarose gel electrophoresis visualized the RT-LAMP product, which exhibited multiple bands of increasing molecular weight, as expected[5] (Fig. 1b). We first verified that Cas9 can efficiently cleave RT-LAMP product by mixing diluted

RT-LAMP product with Cas9 in complex with crRNA and tracrRNA (henceforth referred to as gRNA) targeting the SARS-CoV-2 N-gene (Cas9-19n-N2-gRNA#1) (Supplementary Fig. 1), or a non-targeting gRNA. We observed a band shift in RT-LAMP product on agarose gel with the use of Cas9/gRNA targeting the SARS-CoV-2 sequence ('19n') compared to the use of non-targeting Cas9/gRNA ('neg'), suggesting efficient Cas9 cleavage (Fig. 1b). However, addition of $\lambda$-exo to the reaction unexpectedly did not lead to degradation of cleavage product (Fig. 1b). We hypothesized that $\lambda$-exo loading was inhibited by Cas9 molecules remaining on the target DNA after cleavage[18,19]. To overcome this obstacle, we hypothesized that a highly processive helicase, such as the recently engineered Rep-X "super"-helicase[20], could be used to evict Cas9, potentially through loading onto the cleaved nontarget strand[21]. Indeed, we found that addition of Rep-X and ATP to the $\lambda$-exo reaction greatly increased digestion ('19n') (Fig. 1b, c). Cas9 combined with a non-target gRNA ('neg') resulted in no detectable degradation upon addition of Rep-X and/or $\lambda$-exo, consistent with Cas9 cleavage being the actuator for DNA degradation via $\lambda$-exo (Fig. 1b, c).

To greatly simplify the diagnostic assay, we created a "master mix" composed of Glycine–KOH buffer with preformed SpCas9–gRNA complex, $\lambda$-exo, Rep-X, ATP, and $MgCl_2$. Starting from RNA as the sample of interest, RT-LAMP was first employed for isothermal amplification at 65 °C for 30 min on a heat block (alternatively using LAMP for DNA samples of interest) (Fig. 2a). 2 µL of 1:20-diluted RT-LAMP product was incubated with 18 µL of the SENTINEL master mix in two tubes, one composed of on-target (Reaction A) and the other of non-target Cas9/gRNA (Reaction B), for 15–30 min. at room temperature. For a negative control reference to account for variation between reagent batches, RT-LAMP was also performed without any target nucleic acid, followed by the SENTINEL reaction using non-target Cas9/gRNA (Reaction C). Relative DNA concentrations from Reactions A–C were conveniently measured using a fluorescent plate reader after 1:10 dilution with 1× PicoGreen dye, a dsDNA-specific fluorescent dye that exhibits over 1000-fold increase in fluorescence upon dsDNA intercalation[22].

We aimed to summarize the three fluorescence measurements (A–C) as a single numerical score, where a high score would indicate detection of the target nucleic acid, i.e., a positive test. First, we calculated the fractional increase in LAMP product due to the presence of target sample (B/C), since samples can only have a positive test if LAMP successfully amplified. Second, we determined the fractional reduction in dsDNA fluorescence of LAMP product with use of target versus non-targeting Cas9/gRNA (1−A/B). If the target product was amplified by LAMP, target Cas9/gRNA should efficiently cleave to allow $\lambda$-exo degradation, resulting in a larger reduction. In contrast, if undesired, off-target product was amplified by LAMP, then use of target Cas9/gRNA would have minimal effect relative to a non-targeting Cas9/gRNA, resulting in no reduction. Both quantities, (1−A/B) and (B/C), should have large values to have a positive test, motivating the final SENTINEL score to be computed as (1−A/B)*(B/C). The SENTINEL score is high if LAMP results in amplification products and if these products are degraded by the nucleases, and is low if either condition is not met (Supplementary Fig. 2a). Perturbation analysis varying either C or A relative to B (% reduction) illustrates how the score's value may vary, but the difference in scores between positive and negative samples remains separable and directly related to the extent of their respective $\lambda$-exo mediated degradation (Supplementary Fig. 2b, c).

With the assay and its performance metric at hand, we evaluated the compatibility of the assay for different input types, as well as its limit of detection. Using SARS-CoV-2 in vitro transcribed single-stranded RNA (ssRNA) or a double-stranded DNA

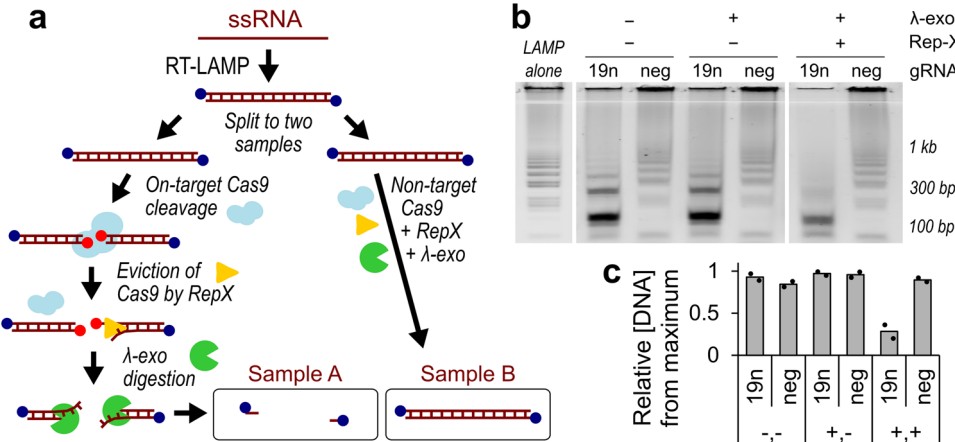

**Fig. 1 Schematic and characterization of endonuclease actuated nucleic acid detection. a** Schematic of SENTINEL mechanism of action. Blue circles at the 5′ ends of the RT-LAMP product represent 5′ phosphorothiolate modifications. Red circles at the 5′ ends represent 5′ phosphates exposed from endonuclease action. **b** Agarose gel electrophoresis of (lane 1) SARS-CoV-2 N-gene RT-LAMP product alone, or with additional (lane 2, 3) Cas9 cleavage using on-target (19n) and non-target (neg) gRNA, (lane 4, 5) digestion with λ-exo, and (lane 6, 7) unwinding with RepX superhelicase. The RT-LAMP product (lane 1) is not altered with exposure to λ-exo (lane 5—'neg'), suggesting that 5′ phosphorothiolate effectively prevents λ-exo end-degradation. Lane 1–7 is oriented from left to right. Lane 2–3: On-target gRNA led to downward shift in gel bands, consistent with cleavage of RT-LAMP product. Lane 4–5: Addition of λ-exo did not lead to appreciable change in cleaved RT-LAMP product. Lane 6–7: Addition of λ-exo with Rep-X led to appreciable degradation of RT-LAMP product, only for the sample with on-target gRNA. In contrast, the sample with off-target gRNA was consistent across panel (**b**). **c** Quantification of panel **b** lanes 2–7 across two biological replicates (black dots), by measuring average fractional intensities of each lane on the agarose gel relative to the maximum intensity across all lanes. Lanes 2–7 in panel **b** correspond to lanes 1–6 in this panel, both oriented from left to right.

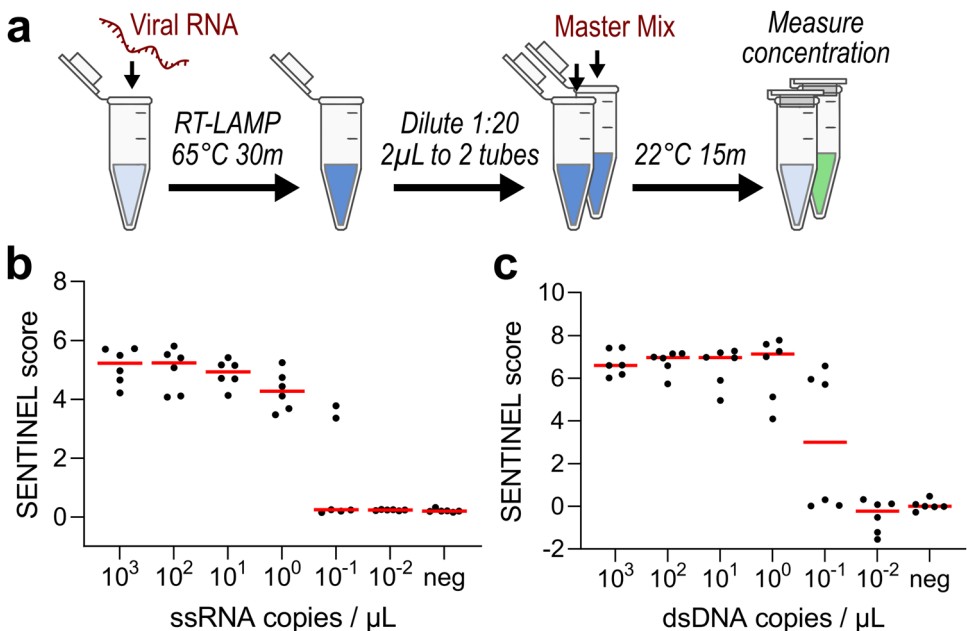

**Fig. 2 Limit of detection for endonuclease-actuated nucleic acid detection. a** Schematic of SENTINEL protocol, starting from synthetic, in vitro transcribed viral RNA. **b** and **c** Limit of detection on **b** ssRNA and **c** dsDNA input. Red lines correspond to median of six experimental replicates for each dilution.

(dsDNA) plasmid encoding the N-gene, we performed (RT)-LAMP using 5′ phosphorothiolated LAMP primers, followed by incubation with SENTINEL master mix containing either on-target gRNA targeting the SARS-CoV-2 template sequence or a non-targeting gRNA. We evaluated the capability of the assay to detect varying concentrations of template though sequential serial dilutions. First, we determined that the master mix was functional-input concentrations above 1 copy per microliter resulted in average SENTINEL scores of over 4, versus a score that approaches 0 when no template was supplied or template quantity under the limit of detection (Fig. 2b, c). Furthermore, we

found that this strategy could reliably detect down to approximately 1 ssRNA or dsDNA molecules per microliter of input sample, which is in the attomolar concentration range. The sensitivity of this assay is bounded by the sensitivity of the LAMP step, which has been shown to be comparable to gold standard quantitative PCR (qPCR)[23]. Together, these results demonstrate that our assay exhibits attomolar levels of sensitivity with clear discrimination between positive and negative test results.

Next, we evaluated the assay's ability to distinguish between closely related target sequences, the stability of its reagents, and its required reaction duration. We performed the assay on in vitro

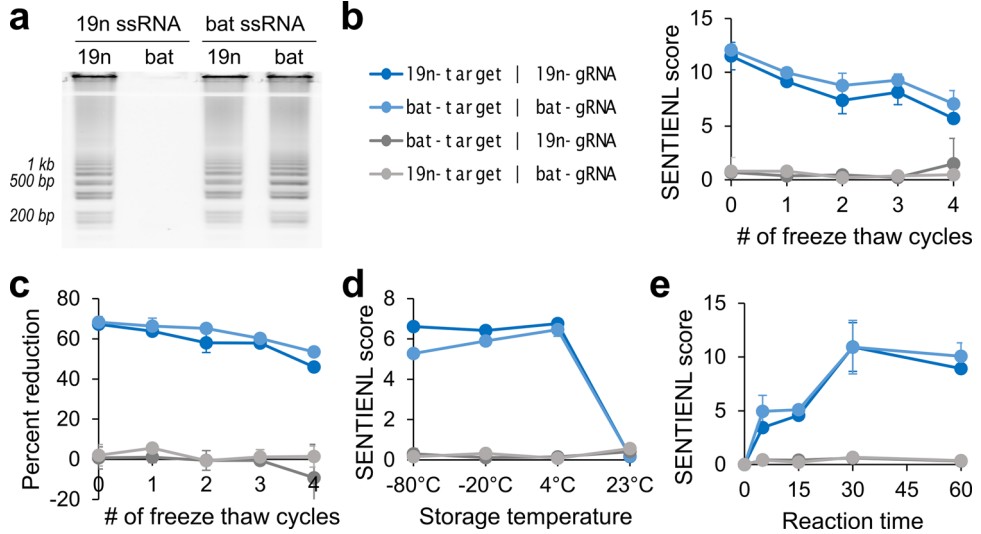

**Fig. 3 Reagent stability, reaction duration, and distinguishing between related sequences. a** Result of isothermal amplification (RT-LAMP) alone for nucleic acid detection. Agarose gel of RT-LAMP product with use of either SARS-CoV-2 (19n) or bat-SL-CoVZC45 (bat) RT-LAMP primers for detection of 19n ssRNA or bat ssRNA, respectively. Notably, detection for SARS-CoV-2 (using 19n RT-LAMP primers) resulted in a false positive when bat ssRNA was in the sample. **b** and **c** Stability of SENTINEL master mix to freeze–thaw cycles, measured with (**b**) SENTINEL scores or with (**c**) percent reduction in DNA concentrations with use of the on-target versus non-targeting gRNA. The legend is the same for panels (**b**)–(**e**): 19n-target and bat-target indicate SARS-CoV-2 and bat-SL-CoVZC45 ssRNA samples, respectively, while 19n-gRNA and bat-gRNA indicate Cas9/gRNA targeting SARS-CoV-2 and bat-SL-CoVZC45 sequences, respectively. **d** Stability of SENTINEL master mix to 1-month storage in −80, −20, 4, or 23 °C (room temperature). The master mix only loses activity after 1-month storage in room temperature. **e** SENTINEL score as a function of room temperature reaction time. **b**–**e** All error bars represent ±1 standard deviation from mean, from three experimental replicates.

transcribed synthetic RNA encoding either SARS-CoV-2 (19nCoV) or the closely related bat-SL-CoVZC45 (batCoV)[24] N-gene. For the SENTINEL reaction, they were paired with Cas9 gRNAs targeting either 19nCoV (Cas9-19n-N2-gRNA#1) or batCoV (Cas9-bat-N2-gRNA#1), whose sequences differed by 5 nucleotides (Supplementary Fig. 1). Notably, use of RT-LAMP alone for 19nCoV detection resulted in a false positive readout when batCoV ssRNA was in the sample solution (Fig. 3a). In comparison, our method did not have a false positive readout–correct pairings of RT-LAMP product with gRNA resulted in a score of over 6, while incorrect pairings (i.e., 19nCoV gRNA with batCoV RT-LAMP product) resulted in a SENTINEL score of under 2 (Fig. 3b). We verified that this difference was due to large, expected reductions in DNA concentrations with the use of the on-target versus non-targeting gRNA, as measured by PicoGreen fluorescence (Fig. 3c). In addition, the master mix was stable for multiple −80 °C to room temperature freeze–thaw cycles (Fig. 3b, c), as well as long-term storage up to 1 month in −80, −20, and 4 °C, but not room temperature (Fig. 3d). Next, by performing the endonuclease reaction as a function of time, we found that even 5 min was sufficient to achieve clear discrimination between expected positive and negative samples (Fig. 3e).

Because SENTINEL relies on targeted endonuclease cleavage to actuate exonuclease-driven target discrimination, we evaluated whether other CRISPR endonucleases such as enhanced-specificity Cas9s and Cas12a[25] would be compatible. To better evaluate the specificity advantages of engineered Cas9s in the SENTINEL system, we designed new gRNAs targeting the N-gene of SARS-CoV-2 (Cas9-19n-N2-gRNA#2) and bat-SL-CoVZC45 (Cas9-19n-N2-gRNA#2) that differed by only two nucleotides in the protospacer (Supplementary Fig. 1). Use of wild type Cas9 for 19nCoV detection was notable for a false positive readout when batCoV ssRNA was in the sample solution (Fig. 4a). In contrast, use of enhanced specificity HiFi Cas9 and eSpCas9[26,27] resulted in a much lower SENTINEL score for incorrect pairings of input

nucleic acids and gRNA, consistent with better discrimination of two-nucleotide mismatches using enhanced specificity Cas9 systems (Fig. 4a–c). We also determined that use of Cas12a (AsCpf1) with gRNAs targeting either the 19nCoV (Cpf1-19n-N2-gRNA#1) or batCoV (Cpf1-bat-N2-gRNA#1) N-gene, which differed by 5 nucleotides in the protospacer (Supplementary Fig. 1), resulted in assay performance comparable to that of Cas9 (Fig. 4d). Notably, Rep-X and ATP were dispensable for the Cas12a reaction, consistent with automatic Cas12a departure from target DNA after cleavage[28], which would facilitate λ-exo loading and target DNA degradation without need for Rep-X.

We also tested the capability of restriction enzymes[29] to act as the sequence-specific endonuclease to actuate λ-exo cleavage. We identified the 6-cutter restriction enzyme AfeI to have one target site in the 19nCoV RT-LAMP product, but not in the batCoV product due to a single mismatch. We replaced the previous Cas9/gRNA and Rep-X components with AfeI, then performed SENTINEL (30 min at room temperature) on the RT-LAMP products of samples containing either 19nCoV or batCoV ssRNA. Because the majority of restriction enzymes are known to be functional in the acetate-based CutSmart buffer, we evaluated the reaction using that buffer as well as the previous Glycine–KOH-based λ-exo buffer. For both CutSmart and λ-exo buffer, we verified positive signal only for the 19nCoV sample, but not for the batCoV sample (Fig. 4e). The SENTINEL score was higher using AfeI compared to CRISPR enzymes, due to a modest increase in degradation efficiency (1−A/B) and a greater increase in the ratio of B/C attributed to reduced background fluorescence (Supplementary Fig. 3a, b). Together, these results demonstrate that the assay has a high degree of flexibility for endonuclease selection and high specificity.

Next, we determined the ability of this assay to detect SARS-CoV-2 virus particles in human saliva. We obtained heat-inactivated SARS-CoV-2 virus originating from the culture medium of SARS-CoV-2-infected VeroTMPRSS2 cells[30]. To mimic detection of viral particles in patient samples, we diluted virus titer in viral transport media (VTM), followed by 1:4

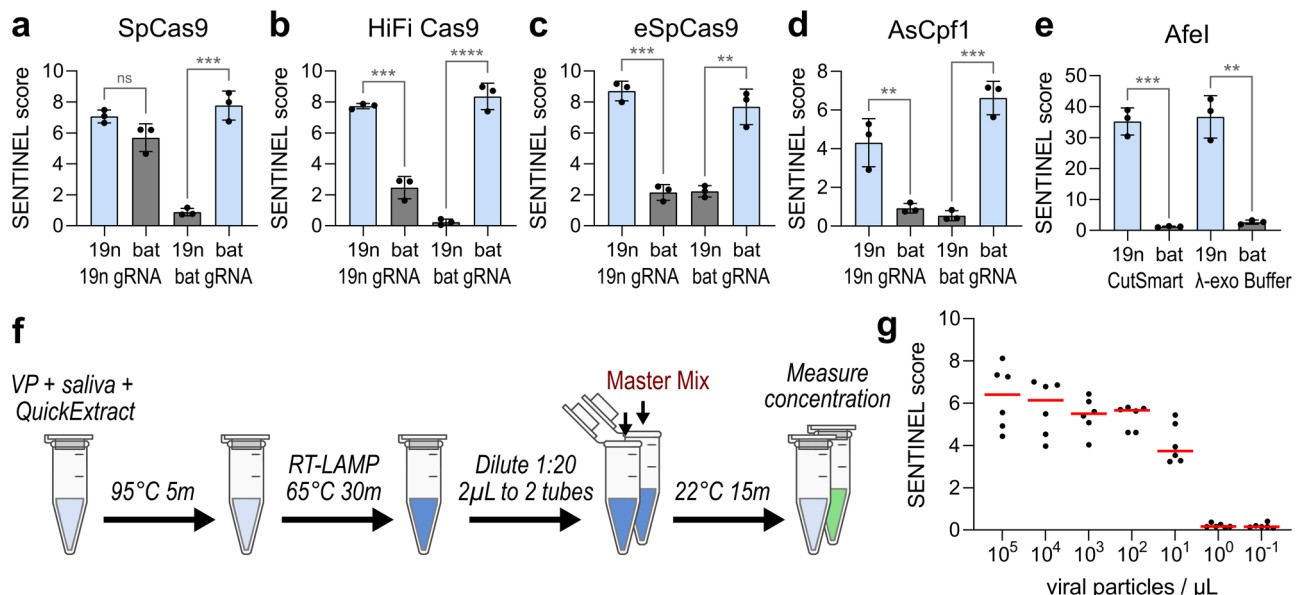

**Fig. 4 Compatibility with different sequence-specific endonucleases and with viral particles in human saliva. a–d** Use of SENTINEL to detect the N-gene of SARS-CoV-2 (19n) vs. bat-SL-CoVZC45 (bat), using guide RNA targeting the N-gene of SARS-CoV-2 (19n gRNA) vs. bat-SL-CoVZC45 (bat gRNA). **a** Wild type SpCas9, **b** HiFi Cas9, **c** eSpCas9, and **d** AsCpf1 were used as the endonuclease. Light blue bars indicate expected positive samples, while gray bars indicate expected negative samples. **e** Use of AfeI as the endonuclease to discriminate between a single-nucleotide difference in sequence between the N-gene of SARS-CoV-2 (19n) and bat-SL-CoVZC45 (bat). CutSmart, and the λ-exo buffer used with other endonucleases, were both compatible. **a–e** All error bars represent ±1 standard deviation from mean, from three experimental replicates plotted as round data points. n.s. indicates not significant, * indicates $p < 0.05$, ** indicates $p < 0.01$, and *** indicates $p < 0.001$ from Student's $t$-test. **f** Schematic of SARS-CoV-2 viral particle detection from human saliva using SENTINEL. **g** Limit of detection from serial dilutions of viral particles in viral transport media, which was mixed into saliva before subject to the SENTINEL assay. Red lines correspond to median of six experimental replicates.

addition into mixed human saliva. To lyse the virus and extract its RNA, we adapted a previous protocol[31] by adding 1:1 of QuickExtract lysis buffer, followed by heating to 95 °C for 5 min. 1 μL of lysed viral particles was used for RT-LAMP and subsequent steps of the SENTINEL assay (Fig. 4f). By evaluating serial dilutions of the virus titer in saliva, we observed high sensitivity, consistently detecting down to 10 particles per microliter of the original input (Fig. 4g).

Finally, we evaluated the ability for SENTINEL to detect SARS-CoV-2 virus from blinded nasopharyngeal (NP) swabs of 50 patients, 25 of which had detectable quantities of SARS-CoV-2 virus from quantitative PCR (qPCR)[32,33]. We directly mixed 2.5 μL of patient sample with 2.5 μL of QuickExtract Lysis buffer, heated to 95 °C for 8 min, then directly proceeded with the SENTINEL assay (Fig. 5a). For NP swabs with qPCR-detectable SARS-CoV-2, higher SENTINEL scores corresponded to greater quantity of target sequence, i.e., lower qPCR quantification cycle (Cq) (Fig. 5b). SENTINEL exhibited a sensitivity of 0.72 for this blinded set of patient samples, sensitivity of 1.00 (no false negatives) for the subset of samples with Cq under 30 ($p < 0.001$), and specificity of 1.00 (no false positives) (Fig. 5b, c). Receiver operating characteristic (ROC) analysis compared to qPCR Cq showed superior performance of SENTINEL over RT-LAMP alone (Fig. 5d), due to potential for non-specific amplification from RT-LAMP (Supp. Fig. 3c, d)[5–12]. Together these results demonstrate the applicability of SENTINEL for rapid detection of SARS-CoV-2 virus directly from patient-derived samples, and improvement over isothermal amplification (RT-LAMP) alone by eliminating potential false positives derived from non-specific amplification.

## Discussion

SENTINEL is a first-in-class method for nucleic acid detection that utilizes the target cleavage properties of sequence-specific endonucleases to verify sequence identity. Our approach is generalizable, in that any sequence-specific endonuclease can be utilized, and orthogonal to existing technologies by using reagents, such as lambda exonuclease and Rep-X, not previously used for nucleic acid detection. These features also allow SENTINEL to be rapidly deployed to meet the demands of diagnostic nucleic acid testing, which, as demonstrated by the recent COVID-19 pandemic, is greatly expedited by the availability of diverse assay methodologies that can overcome supply chain bottlenecks[3]. Furthermore, the limited number of liquid-handling steps and use of plate readers for readout allow SENTINEL to be amenable to 96/384-well plate formats and automation using liquid-handling robots for high-throughput sample processing. Incorporation of rapid RNA cleanup methods, such as using magnetic beads, may further improve detection sensitivity[34]. Future directions include extension to other formats such as portable diagnostic devices[35] and further optimization of the protocol for patient-derived samples to achieve clinical-grade diagnostic testing for COVID-19 using SENTINEL. The emergence of new COVID-19 variants may present new challenges to nucleic acid detection with respect to increased false negatives[36], but SENTINEL is poised to address these challenges by leveraging capacity for single nucleotide discrimination to distinguish between different virus variants.

Other CRISPR-based methods using Cas12 (DETECTR) and Cas13 (SHERLOCK) exhibited comparable features to our strategy[13,14]. The original SHERLOCK and DETECTR assays also require at least one heating temperature (37 or 65 °C) for isothermal amplification before CRISPR-based detection, and require ~1 h to complete the entire protocol. Recently, both strategies have been adapted for SARS-CoV-2 detection[31,37]. The recent SHERLOCK assay variant for SARS-CoV-2 detection, STOPCovid.v2, simplified the protocol by designing a one-pot reaction that only requires one heating temperature. Compared to

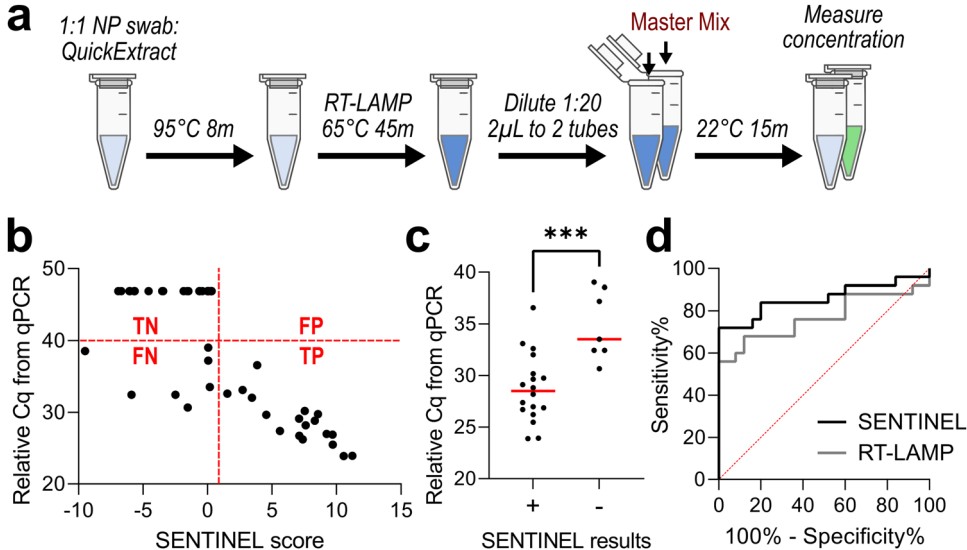

**Fig. 5 SENTINEL on 50 blinded nasopharyngeal swabs from patients. a** Schematic of SARS-CoV-2 viral particle detection using SENTINEL from patient nasopharyngeal swabs in viral transport media (VTM). **b** Graph of SENTINEL score versus Cq from qPCR for the 50 blinded patient samples (25 with detectable virus from qPCR—Cq below 40, 25 negative controls—Cq above 40). qPCR Cqs were initially measured from 300 μL of patient sample, and adjusted here to be consistent with the 2.5 μL of patient samples used for SENTINEL by adding $\log_2(300/2.5) = 6.9$. The four red labels divide the graph into True Positives ('TP'), False Positives ('FP'), True Negatives ('TN'), and False Negatives ('FN'). **c** True positives with SENTINEL ('+') had significantly lower qPCR Cq compared to false negatives ('−') ($p < 0.001$) from Student's $t$-test. **d** Receiver operating characteristic (ROC) curves using SENTINEL versus RT-LAMP alone.

DETECTR for SARS-CoV-2, our strategy requires only 1 versus 2 different heating temperatures (because our second step is performed at room temperature whereas DETECTR's second step is performed at 37 °C). Because DETECTR requires Cas12, only sequence regions adjacent to a Cas12-specific protospacer adjacent motif (PAM) can be evaluated, whereas the flexible choice of endonuclease with SENTINEL allows evaluation of regions next to PAMs of other CRISPR enzymes, or potentially even without PAM restrictions using near-PAMless Cas9 variants[38]. Our design is also related to others that combine target cleavage with isothermal amplification[39,40]. SENTINEL and related methods all have the risk of post-amplification contamination from downstream manipulation of amplification product, which can potentially be resolved using validated strategies such as UV-light irradiation, enzymatic inactivation with uracil-N-glycosylase[41], and newer enzymatic "eraser" strategies[42].

In conclusion, SENTINEL expands the scope of nucleic acid detection by combining the convenience of isothermal amplification with improved specificity to discriminate between one or two base pair differences. It extends nucleic acid detection by utilizing the on-target cleavage properties, rather than bystander collateral cleavage, of CRISPR enzymes to enable use by diverse CRISPR systems from enhanced-specificity Cas9 to Cas12a, as well as other endonucleases such as restriction enzymes. The entire protocol can be completed in under 1 h and only requires a 65 °C heat block and fluorometer on top of basic laboratory materials. The assay master mix is also stable to multiple freeze–thaw cycles and long-term storage.

## Methods

**SpCas9 purification**. BL21-CodonPlus (DE3)-RIL competent cells (Agilent Technologies 230245) were transformed with Cas9 plasmid (Addgene #67881) and inoculated in 25 mL of LB-ampicillin media. The bacteria culture was first allowed to grow overnight (37 °C, 220 rpm) and then transferred to 2 L of LB supplemented with ampicillin and 0.1% glucose until $OD_{600}$ of ~0.5. Subsequently, the cells were induced with IPTG at a final concentration of 0.2 mM and maintained overnight at 18 °C. The bacteria cells were pelleted at $4500 \times g$, 4 °C for 15 min and resuspended in 40 mL of lysis buffer containing 20 mM Tris pH 8.0, 250 mM KCl, 20 mM imidazole, 10% glycerol, 1 mM TCEP, 1 mM PMSF, and cOmplete™ EDTA-free

protease inhibitor tablet (Sigma-Aldrich 11836170001). This cell suspension was lysed using a microfluidizer and the supernatant containing Cas9 protein was clarified by spinning down cell debris at $16,000 \times g$, 4 °C for 40 min and filtering with 0.2 μM syringe filters (Thermo Scientific™ F25006). 4 mL Ni-NTA agarose bead slurry (Qiagen 30210) was pre-equilibrated with lysis buffer. The clarified supernatant was then loaded at 4 °C. The protein-bound Ni-NTA beads were washed with 40 mL wash buffer containing 20 mM Tris pH 8.0, 800 mM KCl, 20 mM imidazole, 10% glycerol, and 1 mM TCEP. Gradient elution was performed with buffer containing 20 mM HEPES pH 8.0, 500 mM KCl, 10% glycerol, and varying concentrations of imidazole (100, 150, 200, and 250 mM) at 7 mL collection volume per fraction. The eluted fractions were tested on an SDS–PAGE gel and imaged by Coomassie blue (Bio-Rad 1610400) staining. To remove any DNA contamination, 5 mL HiTrap Q HP (Cytiva 17115401) was charged with 1 M KCl and then equilibrated with elution buffer containing 250 mM imidazole. The purified protein solution was then passed over the Q column at 4 °C. The flow-through was collected and dialyzed in a 10 kDa SnakeSkin™ dialysis tubing (Thermo Fisher 68100) against 1 L of dialysis buffer (20 mM HEPES pH 7.5, 500 mM KCl, 20% glycerol) at 4 °C, overnight. Next day, the protein was dialyzed for an additional 3 h. in fresh 1 L of dialysis buffer. The final Cas9 protein was concentrated to 10 μg/μL using Amicon Ultra-15 Centrifugal Filter Unit, Ultracel-10 (Millipore Sigma UFC901008), aliquoted, and flash-frozen and stored at −80 °C.

**eSpCas9 purification**. The purification protocol was adapted from the manuscript associated with the eSpCas9 plasmid (Addgene #126769). Briefly, the eSpCas9 plasmid was transformed into BL21 Rosetta 2 (DE3) cells (Millipore EMD 71397) then grown in LB media with 10 μg/mL Kanamycin overnight at 37 °C. 10 mL of this culture was inoculated into 1 L of LB media with 10 μg/mL Kanamycin and grown to a final cell density of 0.6 $OD_{600}$, then chilled at 18 . The protein was expressed at 18 °C for 16 h. following induction with 0.2 mM IPTG. The cells were centrifuged at $6000 \times g$ for 15 min at 4 °C, resuspended in 30 mL of lysis buffer (40 mM Tris pH 8.0, 500 mM NaCl, 20 mM imidazole, 1 mM TCEP) supplemented with cOmplete™ EDTA-free protease inhibitor tablet (1 tablet/30 mL; Sigma-Aldrich 11836170001), and then sonicated on ice. The lysate was cleared by centrifugation at $48,000 \times g$ for 40 min at 4 °C, which was then bound to a 5 mL Mini Nuvia IMAC Ni-Charged column (Bio-Rad 7800812). The resin was washed extensively with a solution of 40 mM Tris pH 8.0, 500 mM NaCl, 20 mM imidazole, and the bound protein was eluted by a solution of 40 mM Tris pH 8.0, 250 mM imidazole, 150 mM NaCl, 1 mM TCEP. 10% glycerol was added to the eluted sample and the His6-MBP fusion protein was cleaved by TEV protease (Addgene pRK793) (3 h at 25 °C). The volume of the protein solution was made up to 100 mL with buffer (20 mM HEPES pH 7.5, 100 mM KCl, 1 mM DTT). The cleaved protein was purified on a 5 mL HiTrap SP HP cation exchange column (GE Healthcare 17115201) and eluted with 1 M KCl, 20 mM HEPES pH 7.5, 1 mM DTT. The protein was further purified by size exclusion chromatography on a HiPrep 26/60 Sephacryl S-200-HR column (GE 17-1195-01) in 20 mM HEPES pH 7.5, 200 mM

KCl, 1 mM DTT, and 10% glycerol. The eluted protein was confirmed by SDS–PAGE and SimplyBlue™ SafeStain (Invitrogen LC6060). The protein was stored at −20 °C.

**Rep-X purification.** *E. coli* Rep helicase was purified and crosslinked as previously described (Arslan et al.[20]). Briefly, the pET28(+) vector containing Rep-DM4 sequence was transformed into E. coli BL21(DE3) cells and grown in TB medium. When optical density reached OD600 = 0.6 the protein overexpression was induced with 0.5 mM IPTG. The cells were incubated overnight at 18 °C and harvested by centrifugation at 10,000 × g. Our Rep-DM4 contains 6x-His tag on its N-terminus for Ni-NTA affinity-column-based purification. The cell pellet was resuspended in a lysis buffer and cells were lysed with a sonicator. After binding Rep to Ni-NTA column and several washes, Rep was eluted with 150 mM imidazole-containing buffer. Rep concentration was kept below 4 mg/mL (~50 μM) to avoid aggregation.

Our Rep-DM4 mutant has four native cysteines removed (C18L, C43S, C167V, C612A), while C178 is kept and S400C mutation is introduced for crosslinking. C178 and C400 are linked with bismaleimidoethane crosslinker (BMOE) to lock Rep in a closed conformation and form Rep-X. Optimal crosslinking is achieved at Rep concentration between 20 and 25 μM, with Rep to BMOE ratio of 1:5. Excess imidazole and crosslinker are removed by overnight dialysis in the storage buffer (50% glycerol, 600 mM NaCl, 50 mM Tris, pH 7.6). The samples are stored at −80 °C. This method achieves nearly 95% crosslinking efficiency.

**Generating synthetic ssRNA via in vitro transcription.** For SARS-CoV-2 N-gene, perform PCR using 0.5 μL each of 10 μM forward and reverse primers (IVT_19n_N2_FWD, IVT_19n_N2_REV; primer sequences in Table S1). Mix with 4 μL of water, 0.5 μL of 2019-nCoV_N_Positive Control template plasmid (Integrated DNA technologies 10006625), and 10 μL of Q5 2× master mix (New England BioLabs M0494). Thermocycle at 98 °C for 30 s for initial denaturation, followed by 35 cycles of 98 °C for 10 s, 69 °C for 20 s, 72 °C for 30 s; then final extension of 72 °C for 2 min and 4 °C hold. Use QIAquick PCR purification kit (Qiagen 28104) to clean up PCR reaction and elute in 35 μl EB supplied with kit—agarose gel of PCR product should visualize band at 931 base pairs.

For in vitro transcription, use the HiScribe T7 kit (New England BioLabs E2050) by mixing 17 μL of purified PCR product with 3 μL of 10× reaction buffer, 8 μL of NTP mix, 2.5 μL of 50 mM DTT, and 2 μL of T7 RNA polymerase mix for a 30 μL total volume. Incubate at 37 °C at least 2 h. Then, add 20 μL of water and 1 μL of DNase I, mix, and incubate for another 37 °C for 15 min. Perform reaction cleanup using Monarch® RNA Cleanup Kit (New England BioLabs T2040). Measure RNA concentration using Nanodrop.

For bat-SL-CoVZC45 N-gene, perform the same protocol with IVT_bat_N2_FWD, IVT_bat_N2_REV primers (sequences in Table S1), with the SARS-CoV Control template plasmid (Integrated DNA technologies 10006624).

**Making stock reagents (LAMP primer mix, cr/tracrRNA, buffers, and master mix) for CRISPR-based SENTINEL assay.** Make 100 μL (RT)-LAMP primer mix by mixing 8 μL of FIP, 8 μL of BIP, 2 μL of F3, 2 μL of B3, 4 μL of LF, 4 μL of LB, and 56 μL of nuclease free water. All primers stocks are 100 μM in TE buffer, and sequences in Table S1.

Make stock 2× SENTINEL buffer by mixing 1-part 10× lambda exonuclease reaction buffer (New England BioLabs M0262), 1 part 100 mM NaCl, 1 part 100 mM MgCl₂, and 2-part 10 mM ATP, resulting in a solution composed of 67 mM Glycine–KOH, 2.5 mM MgCl₂, 50 μg/mL BSA, 2 mM ATP, 10 mM MgCl₂, and 10 mM NaCl.

To make 10 μM cr/tracrRNA, anneal 3 μL of crRNA with 3 μL of tracrRNA (Integrated DNA Technologies), both at stock concentrations of 100 μM in Duplex buffer (Integrated DNA Technologies 11-01-03-01). Heat at 95 °C in a thermocycler with heated lid for 3 min, cool on benchtop for 5 min, then add 27 μL of Duplex buffer to make 30 μL total.

To make 10 μM Cas9, mix 5 μL of 10 μg/μL Cas9 with 25 μL of Dialysis Buffer (20 mM HEPES pH 7.5, 500 mM KCl, 20% glycerol; this is the storage buffer used from Cas9 purification) to make 30 μL total.

To make 10 μM Cpf1, mix 5 μl of 10 μg/μL Alt-R® A.s. Cas12a (Cpf1) Ultra (Integrated DNA Technologies 10001272) with 25 μL of dialysis buffer from Cas9 purification to make 30 μL total (Cpf1 and Cas9 are approximately the same molecular weight).

To make Cas9-based 1× SENTINEL master mix for one reaction, mix 80 μL nuclease free water with 100 μL 2× SENTINEL buffer. Then, add 2 μL of 10 μM annealed cr/trRNA and 1.6 μL of 10 μM Cas9, then mix again. After room temperature incubation for 30 min to form the Cas9–gRNA complex, add 10 μL of 1 μM Rep-X, and 2 μL of Lambda Exonuclease (New England BioLabs M0262), then mix again. This is sufficient for 10 reactions. This master mix can be made in larger batches, aliquoted to smaller volumes, and stored in −20 or −80 °C.

To make Cpf1-based 1× SENTINEL master mix, replace Cas9 with 1.6 μL of 10 μM Cpf1. Rep-X is also not required and is omitted from the master mix.

For experiment with omission of Rep-X and/or Lambda Exonuclease, add equivalent volumes of water instead.

**CRISPR-based SENTINEL assay on synthetic ssRNA or dsDNA.** For a 20 μL reaction, mix 7 μL of nuclease free water, 2 μL of phosphorothiolated LAMP primer mix, and 10 μL of 2× LAMP master mix (New England BioLabs E1700) per reaction. Mix in 1 μL of diluted, synthetic ssRNA or dsDNA, then incubate at 65 °C for 30 min. The reaction can be scaled up to 50 μL. Afterwards, add in TE buffer for 1:20 dilution. Take 2 μL of this diluted (RT)-LAMP product, mix with 18 μl of 1× SENTINEL master mix, then leave at room temperature for 30 min. This is done twice—once with master mix containing on-target gRNA, and another with master mix containing non-target gRNA. Add in 180 μL of 1× PicoGreen solution (1 μL PicoGreen reagent with 200 μL TE buffer) (Thermo Fisher P7589), mix well, then load all to a well of Nunc™ F96 MicroWell™ Black Polystyrene Plate (Thermo Fisher 237105). Using excitation wavelength of 485 nm and emission wavelength of 528 nm, measure the sample fluorescence using a Synergy H1 plate reader (BioTek).

For the separate negative control, (RT)-LAMP of input without spiked-in ssRNA/dsDNA was used in SENTINEL with the non-target gRNA.

**Restriction enzyme (AfeI) SENTINEL assay on synthetic ssRNA or dsDNA.** Only the reaction master mix composition is modified from the CRISPR-based SENTINEL assay. Prepare reaction master mix by mixing 16 μL water, 2 μL CutSmart Buffer (New England BioLabs), 0.2 μL AfeI (New England BioLabs), and 0.2 μL Lambda Exonuclease (New England BioLabs). This can be used to react with 2 μL of diluted (RT)-LAMP product for the assay.

For the non-targeting condition, AfeI is replaced with equal volume of water.

For the negative control, (RT)-LAMP of input without spiked-in ssRNA/dsDNA was used in SENTINEL where AfeI is replaced with equal volume of water.

**CRISPR-based SENTINEL assay on heat-inactivated viral particles.** Gentamicin/Amphotericin B mixture was first made by mixing equal volumes of 50 mg/mL Gentamicin (Sigma-Aldrich G1397) with 250 μg/mL Amphotericin B (Sigma-Aldrich A2942), then filter-sterilized using a 0.22 μm pore size filter unit. Viral transport media (VTM) was prepared by mixing 500 mL of Hanks balanced salt solution (HBSS), 10 mL heat-inactivated FBS (Corning), and 2 mL of filter-sterilized Gentamicin/Amphotericin B mixture.

Supernatant from SARS-CoV-2-infected Vero cell culture with 10E7 viral particles per μL was heated at 65 °C for 1 h for virus inactivation. Serial dilutions were performed using VTM. 1 μL of the dilution was mixed with 4 μL pooled human saliva (Innovative Research IRHUSL5ML) to simulate virus presence in human saliva. This was mixed 1:1 with QuickExtract™ DNA Extraction Solution (Lucigen QE09050), then immediately heated to 95 °C for 5 min. 1 μL of this final solution was used in a 20 μL SENTINEL reaction, with the remainder of the protocol identical to the section "CRISPR-based SENTINEL assay on synthetic ssRNA or dsDNA".

**Patient consent statement and study cohort.** From April 21, 2020 to July 16, 2020, non-hospitalized adults who were self-isolating after receiving a positive NP SARS-CoV-2 real-time reverse-transcription polymerase chain reaction (rRT-PCR) result from the Johns Hopkins Medical Microbiology laboratory were approached for participation by telephone using a verbal consent script. Obtaining signed informed consent form for subjects enrolled in this study was not initially feasible for study staff due to the contagious nature of COVID-19 being studied under this protocol. Instead, the study staff used a consent waver and obtained verbal consent. Inclusion criteria were age ≥18 years, able to receive study materials while remaining in isolation, and able and willing to perform self-collection of specimens. This protocol and verbal consent were approved by the Johns Hopkins University School of Medicine Institutional Review Board (IRB), and are identical to that of Manabe et al. (2020)[32] and Antar et al. (2021)[33]. All procedures were in accordance with the ethical standards of the Helsinki Declaration of the World Medical Association.

**Specimen collection.** Participants were mailed a sample collection kit that included an international air transport association (IATA)-approved biologic sample container as well as sample collection materials and written instructions for sample collection. Participants self-collected mid-turbinate nasal and oropharyngeal (nasal-OP) swabs; both swabs were placed in 3 mL viral transport medium. All samples were immediately placed in the IATA container and stored in the participant's freezer before shipping. Participants self-collected samples on the day they received the collection materials (day 0) and then subsequently on study days 3, 7, 14. On day 14, the participant shipped the collected samples on ice-cold packs to Johns Hopkins University for analysis using an overnight courier service. The procedure was identical to that of Manabe et al. (2020)[32] and Antar et al. (2021)[33].

**Blinding schema.** Individual 1 has information on ground-truth quantitative PCR readout for each patient sample. 25 SARS-CoV-2 positive samples and 25 SARS-CoV-2 negative samples were randomly placed in a freezer box by Individual 1. When Individual 1 was not present, Individual 2 removed the samples from the freezer box and performed the SENTINEL assay. Individual 1 sends the SENTINEL results to Individual 2 for verification.

**SENTINEL assay on patient nasopharyngeal swabs.** 35 µL of patient nasopharyngeal swabs was mixed 1:1 with QuickExtract™ DNA extraction solution (Lucigen QE09050), then immediately heated to 95 °C for 8 min. 2 µL of this final solution was used in a 50 µL SENTINEL reaction. 45 min was used for the RT-LAMP step, and detection was performed on the SpectraMax i3x plate reader (Molecular Devices). Everything else was identical to the section "CRISPR-based SENTINEL assay on synthetic ssRNA or dsDNA".

**Computation of SENTINEL score.** Reaction A corresponds to the on-target SENTINEL reaction (on-target Cas/gRNA or on-target restriction enzyme). Reaction B corresponds to the non-target SENTINEL reaction (non-target Cas9/gRNA or water in the place of restriction enzyme). Reaction C corresponds to the negative control—the sample expected to have no target nucleic acid, exposed to the non-target SENTINEL reaction. A–C will be the results of Reactions A–C, respectively, as fluorescence measurements (in arbitrary units) on a plate reader.

The first part of the SENTINEL score is fractional reduction in fluorescence with on-target versus non-target Cas9/gRNA, i.e., $1 - A/B$. Next, this value is scaled by the fractional increase in DNA quantity due to LAMP amplification ($B/C$). Together, the SENTINEL score is computed using the following formula: $(1 - A/B) * (B/C)$.

**Statistics and reproducibility.** All statistical analyses were conducted on either Microsoft Excel or GraphPad Prism 9. All data for statistical analysis were numerical, and a sample size and/or replicates of at least 3 were used for hypothesis testing using Student's $t$-test. Replicates were all 'biological', in that that each sample of the replicate was independently conducted and in separate test tubes.

**Reporting summary.** Further information on research design is available in the Nature Research Reporting Summary linked to this article.

## Data availability

Source data underlying main figures are presented in Supplementary Data 1. Uncropped versions of gels and blots are presented in Supplementary Fig. 4.

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

## Acknowledgements

We thank all members of the Ha, Myong, and Manabe labs for comments and suggestions. This work was supported by the JHU COVID-19 Research and Response Program Fund, the Sherrilyn and Ken Fisher Center for Environmental Infectious Diseases Discovery Program, the National Institutes of Health (R35 GM 122569 to T.H., R37 HD 037047 to G.S., U54EB007958-13, U54EB007958-12S1, and CEIRR - N7593021C00045 to Y.C.M., T32 GM 136577 and F30 CA 254160 to R.S.Z.), and the National Science Foundation (MCB 2031094 to T.H.). M.G. was supported by

the NSERC postdoctoral fellowship. T.H. and G.S. are investigators of the Howard Hughes Medical Institute. The content is solely the responsibility of the authors and does not necessarily represent the official views of the National Institutes of Health.

## Author contributions

R.S.Z. and T.H. initially conceived the project. R.S.Z., L.S., and S.C. designed and performed experiments. M.G. purified Rep-X, Y.L. purified SpCas9, D.R. purified eSpCas9 using protocol adapted by D.R. and G.S. A.P. generated heat-inactivated SARS-CoV-2 virus. M.C., J.H., and Y.C.M. obtained patient samples and performed clinical qPCR. All authors contributed to the writing of the manuscript. T.H. supervised the project.

## Competing interests

The authors declare the following competing interests: R.S.Z. and T.H. are listed as authors of a patent application submitted by Johns Hopkins University on the method of nucleic acid detection presented in this manuscript. G.S. serves on the Scientific Advisory Board of Dewpoint Therapeutics, Inc. The remaining authors declare no competing interests.
