## [Peer Review File · Communications Biology]

Reviewers' comments:

Reviewer #1 (Remarks to the Author):

The authors describe their SENTINEL assay, which combines LAMP of SARS-CoV-2 RNA with endonuclease digestion of the amplicon to improve LAMP specificity. The principle justification for this approach is that other nucleic acid detection methods suffer from false-positive results. Yet we know that one of the principle concerns surrounding SARS-CoV-2 diagnostics is the increased proportion of false negative results in nucleic acid-based diagnostics as a result of the emergency of new variants. I don't believe that this work reflects a substantial improvement on SARS-CoV-2 LAMP diagnostics as the authors have not made a compelling case that RT-LAMP specificity is current a cause for concern for SARS-CoV-2 detection.

The gRNAs used are in highly conserved regions of SARS-CoV-2 that overlap with the N2 primer and probe target region of the CDC assay. If the authors are considering that this assay be used in a clinical setting for the detection of SARS-CoV-2 infection, they should consider whether a gRNA that is a little more tolerant to mismatches (i.e. gRNA #1) would actually be preferable should future variants of SARS-CoV-2 have mutations in their target region. Improved specificity is an important consideration in diagnostic design yet so is the likelihood of a clinical sample containing a bat coronavirus.

Introduction

Line 34-36: The authors make the assertion that "Quantitative qPCR...may have difficulty distinguishing between closely related sequences". The opposite is true; probe-based qPCR is exquisitely sensitive when it comes to differentiating single nucleotide differences between related sequences.

Results

Figure 1 should be reorganized so that the panels flow in a more logical fashion. It is incredibly difficult to read. Figure 1b-d has multiple bands visible in the 19n wells, however, it is not stated which band is the correct amplicon size, and it is therefore difficult to determine the efficiency of the degradation as a strong band ~150bp is still present in the 19n reaction in Figure 1d.

Figure S2a: What is the "expected target DNA sequence". Are you referring to the target sequence for the Cas9/gRNA or the target sequence for the LAMP primers? Also, the authors state that the difference in scores is "mainly driven by the difference in A (given B and C are similar)". They should state that it is the difference between A in the positive and negative reactions, and that B and C are similar in their respective positive and negative reactions. It reads as though the authors are claiming that B and C are similar in the positive reaction, and B and C are similar in the negative reaction.

Line 141: What are "appropriate concentrations of template"?

For Fig 1g-h and Fig 2g, the n is not given in the figure legend or elsewhere in the manuscript. Based on the number of points plotted on the graph, it looks as though triplicates were performed. This is not a high enough sample size to determine the limit of detection when using 0, 10 and 100 copies / reaction.

Figure S3 is incredibly difficult to interpret given the author jump between red and grey plot lines for 19n / bat. How does figure S3f differ from Figure 1I? Do they not both show SARS-CoV-2 RNA with it's corresponding on-target gRNA versus the non-target (bat) gRNA? Why is the room temperature reaction time data given in Fig. 1I as the SENTINEL score yet in Fig. S3f and S3g the data is given as a percent reduction in fluorescence?

The data using the AfeI restriction enzyme is impressive yet the authors do not discuss that their results should a 5-10x increase in the SENTINEL score using the restriction enzyme. It would have been prudent to use the AfeI restriction enzyme with the saliva samples, which may have increased the LOD.

Reviewer #2 (Remarks to the Author):

Generally, the authors fabricated an interesting strategy to improve the specificity of LAMP-based methods in detection of SARS-CoV-2 pathogenic nucleic acids. Specifically, various sequence specific endonucleases, including wild type Cas9, its engineered variants, Cas12a and AfeI, were used to introduce 5' phosphorylated double strand break or nicks, which subsequently digested by the highly processive Lambda exonuclease. In general, this is an interesting study, but there are some major points to be further modified before publishing in Communications Biology.

1. Besides false positive, the other drawback of LAMP method is the post-amplification contamination issue, in which amplified products with high concentration might easily splash or form aerosols during experimental operation, causing pollution to air, reagents, pipettes and the gloves and clothes of the operator. That's why in LAMP reactions, tube lids cannot be opened once the amplification initiated. However, in the protocol provided by the authors, LAMP amplified products has to be diluted and mixed with SENTINEL master mix, which may result in serious contamination for subsequent experiments in practical scenarios. The authors should discuss this issue in the manuscript.

2. I am curious that why not use RPA instead of LAMP. In RPA reactions, the authors may able to premix SENTINEL master mix with rt-RPA supplemented with 5'-OH labeled rt-primer and amplification primers to give a one-pot reaction at 37°C.

3. In the manuscript, readouts of detection have to be calculated by equation $(1-A/B) * (B/C)$, in which 3 different SENTINEL reactions has to be set and performed and measured quantitatively. This is unpractical in real diagnosis. I would expect value B and C to be quite constant since they are, experimentally speaking, negative controls. Can the authors give a simplified numeric score? For example, decreasing rates of reaction A (Slope A) after the addition of SENTINEL master mix.

4. The authors stated that "Furthermore, we found that this strategy could reliably detect down to approximately 100 ssRNA or dsDNA molecules per microliter of input sample, which is in the attomolar concentration range. The sensitivity of this assay is bounded by the sensitivity of the LAMP step, which has been shown to be comparable to gold standard quantitative PCR (qPCR)". When using pure nucleic acids fragments as template, LoD of LAMP reaction can easily reach 10 copies/reaction. However, when viral particles were used as templates, LoD of 10 particles per microliter was observed. The authors should explain possible reasons for this phenomenon, and provide representative photographs of LAMP results in SI.

Minor problems:

1. Labels a-l in figure 1 is not in proper order.
2. In Line 157 Figure 1j is wrong.
3. Suggestion: Freeze-thaw property of SENTINEL can be rearranged as Fig3.

Reviewers' comments:

Reviewer #1 (Remarks to the Author):

The authors describe their SENTINEL assay, which combines LAMP of SARS-CoV-2 RNA with endonuclease digestion of the amplicon to improve LAMP specificity. The principle justification for this approach is that other nucleic acid detection methods suffer from false-positive results. Yet we know that one of the principle concerns surrounding SARS-CoV-2 diagnostics is the increased proportion of false negative results in nucleic acid-based diagnostics as a result of the emergency of new variants. I don't believe that this work reflects a substantial improvement on SARS-CoV-2 LAMP diagnostics as the authors have not made a compelling case that RT-LAMP specificity is current a cause for concern for SARS-CoV-2 detection.

Response: We thank the reviewer for a detailed reading of our manuscript and very constructive comments that greatly strengthen our study. The main goal of this study is to present a conceptually novel approach for nucleic acid detection; we used SARS-CoV-2 as proof of concept to demonstrate features and advantages of our approach. One proposed advantage of our system is to avoid the false-positive outcomes, and we agree that in our original submission, we did not present direct evidence of such in our example use cases. In our revision, we collaborated with a clinical laboratory who provided 50 blinded nasopharyngeal (NP) swabs from patient suspected to have COVID-19, and performed SENTINEL versus RT-LAMP alone on these samples. Consistent with literature (references 6 to 12 in the revised text), we observed examples of false positive amplification using RT-LAMP alone, whereas SENTINEL accurately discarded putative false-positives, resulting in improved performance compared to LAMP. These results are featured in the new Figure 5 of our revised manuscript (see below).

Figure 5: SENTINEL on 50 blinded nasopharyngeal swabs from patients

- Schematic of SARS-CoV-2 viral particle detection using SENTINEL from patient nasopharyngeal swabs in viral transport media (VTM).
- Graph of SENTINEL score versus Cq from qPCR for the 50 blinded patient samples (25 with detectable virus from qPCR – Cq below 40, 25 negative controls – Cq above 40). qPCR Cqs were initially measured from 300 µL of patient sample, and adjusted here to be consistent with the 2.5 µL of patient samples used for SENTINEL by adding $\log_2(300/2.5) = 6.9$. The four red labels divide the graph into True Positives ('TP'), False Positives ('FP'), True Negatives ('TN'), and False Negatives ('FN').
- True positives with SENTINEL ('+') had significantly lower qPCR Cq compared to false negatives ('-') (p < 0.001) from Student's t-test.
- Receiver operating characteristic (ROC) curves using SENTINEL versus RT-LAMP alone.

The gRNAs used are in highly conserved regions of SARS-CoV-2 that overlap with the N2 primer and probe target region of the CDC assay. If the authors are considering that this assay be used in a clinical setting for the detection of SARS-CoV-2 infection, they should consider whether a gRNA that is a little more tolerant to mismatches (i.e. gRNA #1) would actually be preferable should future variants of SARS-CoV-2 have mutations in their target region. Improved specificity is an important consideration in diagnostic design yet so is the likelihood of a clinical sample containing a bat coronavirus.

Response: We thank the reviewer for this insightful comment and agree that there are many considerations that go into selecting optimal sequences for clinical testing. Our revision manuscript presents data from patient samples that further validate the method for clinical testing, but there is still an extensive path towards approval in the clinical setting. A test with improved convenience and specificity such as SENTINEL is poised to better distinguish between existing SARS-CoV-2 variants as a screening tool, but high-throughput sequencing would be the preferred strategy for identifying new variants circulating in the community.

Introduction

Line 34-36: The authors make the assertion that “Quantitative qPCR...may have difficulty distinguishing between closely related sequences”. The opposite is true; probe-based qPCR is exquisitely sensitive when it comes to differentiating single nucleotide differences between related sequences.

Response: We thank the reviewer for this correct comment. We have removed the previous assertion.

Results

Figure 1 should be reorganized so that the panels flow in a more logical fashion. It is incredibly difficult to read. Figure 1b-d has multiple bands visible in the 19n wells, however, it is not stated which band is the correct amplicon size, and it is therefore difficult to determine the efficiency of the degradation as a strong band ~150bp is still present in the 19n reaction in Figure 1d.

Response: We agree with the reviewer that Figure 1 was indeed very disorganized. Figure 1 is now split into three separate figures (Figures 1 to 3) to improve clarity and logical flow.

We realized that we failed to indicate what RT-LAMP product normally looks like for the general readership. Figure 1b now shows the product from RT-LAMP alone, which reveals multiple bands of increasing molecular weight, as expected. The sequence of these products is highly repetitive and contain multiple sites of Cas9 cleavage. Therefore, the strong ~150bp band only appears after Cas9 digestion, suggesting that Cas9 efficiently breaks apart large molecular weight DNA into 150bp units. The degradation efficiency using lambda exo is not 100%, as seen by the residual 150bp band (19n) in the new Figure 1e that is significantly less intense than the 19n bands in the new Figures 1c and 1d. The DNA intensities in Figures 1b-e are also quantified in the new Figure 1f. The new Figure 1 is shown below.

Figure 1. Schematic and characterization of endonuclease actuated nucleic acid detection

a) Schematic of SENTINEL mechanism of action. Blue circles at the 5' ends of the RT-LAMP product represent 5' phosphorothiolate modifications. Red circles at the 5' ends represent 5' phosphates exposed from endonuclease action.

b-e) Agarose gel electrophoresis of (b) SARS-CoV-2 N-gene RT-LAMP product alone, or with additional (c) Cas9 cleavage using on-target (19n) and non-target (neg) gRNA, (d) digestion with λ -exo, and (e) unwinding with RepX superhelicase. The RT-LAMP product (panel b) is not altered with exposure to λ -exo ('neg' of panel d), suggesting that 5' phosphorothiolate effectively prevents λ -exo end-degradation.

c) On-target gRNA led to downward shift in gel bands, consistent with cleavage of RT-LAMP product.

d) Addition of λ -exo did not lead to appreciable change in cleaved RT-LAMP product.

e) Addition of λ -exo with Rep-X led to significant degradation of RT-LAMP product, only for the sample with on-target gRNA. In contrast, the sample with off-target gRNA was consistent across panels c-e.

f) Quantification of panels b-d across two biological replicates, by measuring fractional intensities of each lane on the agarose gel relative to the maximum intensity across all lanes. From left to right, the first two lanes quantify panel c, second two lanes quantify panel d, and last two lanes quantify panel e.

Figure S2a: What is the “expected target DNA sequence”. Are you referring to the target sequence for the Cas9/gRNA or the target sequence for the LAMP primers? Also, the authors state that the difference in scores is “mainly driven by the difference in A (given B and C are similar)”. They should state that it is the difference between A in the positive and negative reactions, and that B and C are similar in their respective positive and negative reactions. It reads as though the authors are claiming that B and C are similar in the positive reaction, and B and C are similar in the negative reaction.

Response: We thank the reviewer for the careful reading of this manuscript and constructive comment. We see the ambiguity in our original text; what we meant to say is that both + and – have positive LAMP amplification, but + has a site cleavable by Cas9/gRNA whereas – does not. We have modified the text in Figure S2a to appropriately convey this information. We have also modified the second part of Figure S2a as this reviewer has helpfully suggested to avoid further ambiguities.

Line 141: What are “appropriate concentrations of template”? For Fig 1g-h and Fig 2g, the n is not given in the figure legend or elsewhere in the manuscript. Based on the number of points plotted on the graph, it looks as though triplicates were performed. This is not a high enough sample size to determine the limit of detection when using 0, 10 and 100 copies / reaction.

Response: We thank the reviewer for this comment. In this revision, we have clarified “appropriate concentrations of template” in the text as “input concentrations above 1 copy per microliter”. Furthermore, we repeated the experiment using an optimized protocol with larger reaction volumes, which resulted in improved limits of detections. We also used 6 replicates for each sample, ensuring a larger sample size to better determine limit of detection.

Figure S3 is incredibly difficult to interpret given the author jump between red and grey plot lines for 19n / bat. How does figure S3f differ from Figure 1l? Do they not both show SARS-CoV-2 RNA with it’s corresponding on-target gRNA versus the non-target (bat) gRNA? Why is the room temperature reaction time data given in Fig. 1l as the SENTINEL score yet in Fig. S3f and S3g the data is given as a percent reduction in fluorescence?

Response: We thank the reviewer for this detailed comment. We agree that the previous version of the figures is very difficult to interpret – we believe we have greatly improved the clarity of our figures in this revision. The many red and grey line plots have been combined with improved organization; the new version of these plots can be found in the new Figure 3 (shown below).

Figure 3. Reagent stability, reaction duration, and distinguishing between related sequences

- a) Result of isothermal amplification (RT-LAMP) alone for nucleic acid detection. Agarose gel of RT-LAMP product with use of either SARS-CoV-2 (19n) or bat-SL-CoVZC45 (bat) RT-LAMP primers for detection of 19n ssRNA or bat ssRNA, respectively. Notably, detection for SARS-CoV-2 (using 19n RT-LAMP primers) resulted in a false positive when bat ssRNA was in the sample.
- b) Legend for panels c-f. 19n-target and bat-target indicate SARS-CoV-2 and bat-SL-CoVZC45 ssRNA samples, respectively. 19n-gRNA and bat-gRNA indicate Cas9/gRNA targeting SARS-CoV-2 and bat-SL-CoVZC45 sequences, respectively.
- c-d) Stability of SENTINEL master mix to freeze-thaw cycles, measured with (c) SENTINEL scores or with (d) percent reduction in DNA concentrations with use of the on-target versus non-targeting gRNA.
- e) Stability of SENTINEL master mix to 1-month storage in -80 °C, -20 °C, 4 °C, or 23 °C (room temperature). The master mix only loses activity after 1-month storage in room temperature.
- f) SENTINEL score as a function of room temperature reaction time.
- c-f) All error bars represent ± 1 standard deviation from mean, from 3 experimental replicates.

The data using the AfeI restriction enzyme is impressive yet the authors do not discuss that their results should a 5-10x increase in the SENTINEL score using the restriction enzyme. It would have been prudent to use the AfeI restriction enzyme with the saliva samples, which may have increased the LOD.

Response: We investigated the cause for the 5-10 increase in SENTINEL score, and determined the combined effect of increased degradation efficiency (increased 1-A/B) along with reduced background fluorescence (reduced C leading to greatly increased B/C). These details are included in the new Figures S3a-b, and referenced in the main text.

Reviewer #2 (Remarks to the Author):

Generally, the authors fabricated an interesting strategy to improve the specificity of LAMP-based methods in detection of SARS-CoV-2 pathogenic nucleic acids. Specifically, various sequence specific endonucleases, including wild type Cas9, its engineered variants, Cas12a and AfeI, were used to introduce 5' phosphorylated double strand break or nicks, which subsequently digested by the highly processive Lambda Lambda exonuclease. In general, this is an interesting study, but there are some major points to be further modified before publishing in Communications Biology.

Response: We thank the reviewer for the support of our study, and for the many constructive comments detailed below that greatly strengthen our manuscript.

1. Besides false positive, the other drawback of LAMP method is the post-amplification contamination issue, in which amplified products with high concentration might easily splash or form aerosols during experimental operation, causing pollution to air, reagents, pipettes and the gloves and clothes of the operator. That's why in LAMP reactions, tube lids cannot be opened once the amplification initiated. However, in the protocol provided by the authors, LAMP amplified products has to be diluted and mixed with SENTINEL master mix, which may result in serious contamination for subsequent experiments in practical scenarios. The authors should discuss this issue in the manuscript.

Response: The reviewer raises important points about post-amplification contamination. We included a brief discussion about this issue in the manuscript. We note that many related methods, including SHERLOCK and DETECTR, also involve manipulation of isothermally amplified product and risk post-amplification contamination. We list three different strategies for resolving this issue, including UV-light irradiation, enzymatic inactivation with uracil-N-glycosylase, and newer enzymatic "eraser" strategies. New citations that introduce these strategies are also included for reference.

2. I am curious that why not use RPA instead of LAMP. In RPA reactions, the authors may able to premix SENTINEL master mix with rt-RPA supplemented with 5'-OH labeled rt-primer and amplification primers to give a one-pot reaction at 37oC.

Response: SENTINEL may also be compatible with RPA. Empirically, we found LAMP to produce the most robust and specific pre-amplification. Indeed, this may be a reason why related methods such as SHERLOCK and DETECTR also utilize LAMP instead of RPA for SARS-CoV-2 detection. LAMP was also more convenient to obtain, having multiple commercial sources locally in the US (Thermo Fisher, New England BioLabs) compared to RPA, which is only available from a company in the UK. We are currently working on follow-up directions that aim to combine SENTINEL with other isothermal amplification strategies such as RPA and helicase-dependent amplification in a one-pot reaction.

3. In the manuscript, readouts of detection have to be calculated by equation $(1-A/B) * (B/C)$, in which 3 different SENTINEL reactions has to be set and performed and measured quantitatively. This is unpractical in real diagnosis. I would expect value B and C to be quite constant since they are, experimentally speaking, negative controls. Can the authors give a simplified numeric score? For example, decreasing rates of reaction A (Slope A) after the addition of SENTINEL master mix.

Response: We thank the reviewer for this detailed comment and great suggestions. While three measurements are used in the formula, only 2 reactions are needed per sample, which is more practical. As the reviewer notes, Reaction C is a negative control, and the same value can be used across the entire set of experiments. However, Reaction B is not a true negative control, with the value of B/C reporting on the extent of the isothermal amplification step. Only using Reaction A would lead to difficulty distinguishing between appropriate amplification followed by degradation, from lack of amplification. Looking at reaction A slope is very creative and interesting, but based on our understanding, would require repeated measurements of DNA concentration.

4. The authors stated that “Furthermore, we found that this strategy could reliably detect down to approximately 100 ssRNA or dsDNA molecules per microliter of input sample, which is in the attomolar concentration range. The sensitivity of this assay is bounded by the sensitivity of the LAMP step, which has been shown to be comparable to gold standard quantitative PCR (qPCR)”. When using pure nucleic acids fragments as template, LoD of LAMP reaction can easily reach 10 copies/reaction. However, when viral particles were used as templates, LoD of 10 particles per microliter was observed. The authors should explain possible reasons for this phenomenon, and provide representative photographs of LAMP results in SI.

Response: We thank the reviewer for this detailed observation. The reason for the discrepancy was that the initial LoD detection experiments on pure nucleic acid fragment naively used very small (5ul) reaction volumes, which greatly hurts LoD. Following suggestions by another reviewer, we re-performed all LoD experiments with more experimental replicates, and this time taking the opportunity to use greater (50ul) reaction volumes. Across 6 replicates, we now observe LoD down to 1 to 10 molecules per microliter for pure nucleic acid fragments, and LoD of 10 viral particles per microliter. These new results are shown in the new Figures 2 and 4.

Minor problems:

1. Labels a-l in figure 1 is not in proper order.

Response: We thank the reviewer for this detailed observation. Based on feedback from all reviewers, we have reorganized the old Figure 1 into three new figures – Figures 1, 2, and 3. We believe this reorganization greatly improves the order and logical flow.

2. In Line 157 Figure 1j is wrong.

Response: We thank the reviewer for this detailed observation. We recognize that the old Figure 1j was very confusing, and the corresponding figure panel is now improved in this revision.

3. *Suggestion: Freeze-thaw property of SENTINEL can be rearranged as Fig3.*

Response: We thank the reviewer for this great suggestion – the old Figure 1 is now rearranged into three separate figures.

REVIEWERS' COMMENTS:

Reviewer #2 (Remarks to the Author):

The authors have answered all my technical comments, and the quality of the manuscript has been improved. However, there are some points need to be further improved to meet the requirements of the journal.

1. Figure labels needs to be revised. For example, it is very wired that figure labels 1b-e are presented in individual gel lanes. Also, Fig3b is a figure legend, which is also unusual. Furthermore, the arrangement of subfigures needs to be unified. For example, in other figures, subfigures are arranged firstly horizontally and then vertically, but in Fig3, the order is different.

2. Data presented need to be further checked. For example, the red line (averaged value) in fig3b 10^{-1} is apparently wrong.

REVIEWERS' COMMENTS:

Reviewer #1 originally commented on the lack of information about false negatives. Please discuss this issue within the Discussion and note whether future studies are needed to determine the false negative rate of SENTINEL.

We thank the editor for raising this important point. I understand Reviewer #1 did not get an opportunity to evaluate our previous revision, where we included extensive data on blinded patient samples. Figure 5, which discusses SENTINEL on blinded clinical samples, incorporates data about false negatives by directly shows the samples with false negatives (Figure 5b), and in relation to other classification errors through the ROC curve (Figure 5d). We hope that the editor agrees that Reviewer #1's comment is now appropriately addressed in the Results section without need for further comment in the Discussion.

Reviewer #2 (Remarks to the Author):

The authors have answered all my technical comments, and the quality of the manuscript has been improved. However, there are some points need to be further improved to meet the requirements of the journal.

We thank the reviewer for their strong support of this manuscript.

1. Figure labels needs to be revised. For example, it is very wired that figure labels 1b-e are presented in individual gel lanes. Also, Fig3b is a figure legend, which is also unusual. Furthermore, the arrangement of subfigures needs to be unified. For example, in other figures, subfigures are arranged firstly horizontally and then vertically, but in Fig3, the order is different.

We thank the reviewer for these very helpful comments. Figure 1 now has one label for the entire gel. The figure legend in Figure 3 is now part of panel b. In Figure 3, the arrangement is now changed to horizontally, then vertically, as is standard.

2. Data presented need to be further checked. For example, the red line (averaged value) in fig3b 10^{-1} is apparently wrong.

We thank the reviewer for this comment. We believe the reviewer is referring to Fig 2b. The red line is actually corresponding to the median of 6 experimental replicates, making the red line correct. We verified that the figure legend includes this information. Following the reviewer's great suggestion, we have re-checked all the presented data.